# Correlation of Periodontal Bacteria with Chronic Inflammation Present in Patients with Metabolic Syndrome

**DOI:** 10.3390/biomedicines9111709

**Published:** 2021-11-18

**Authors:** Timea Claudia Ghitea

**Affiliations:** Department of Pharmacy, Faculty of Medicine and Pharmacy, University of Oradea, 1st December Square 10, 410068 Oradea, Romania; timea.ghitea@csud.uoradea.ro

**Keywords:** metabolic syndrome, periodontitis, cytokines, diet therapy

## Abstract

Metabolic syndrome (MS) is correlated with many chronic diseases, and so far is moderately followed and treated. The present study follows a correlation of the presence of pathogens (*Fusobacterium nucleatum, Bacteroides forsythus,* and others) in the gingival crevicular fluid and MS. (1) An important role in the fight against MS is to reduce fat mass, inflammatory mediators, and prevent cytokine-associated diseases. (2) A group of 111 people with MS was studied, divided into 3 groups. The control group (CG) received no treatment for either periodontitis or MS. The diet therapy group (DG) followed a clinical diet therapy specific to MS, and the diet therapy and sports group (DSG) in addition to diet therapy introduced regular physical activity; (3) A statistically significant worsening of periodontopathogens was observed correlated with the advancement of MS (increase in fat mass, visceral fat, and ECW/TBW ratio) in the CG group. In the case of DG and DSG groups, an improvement of the parameters was observed, including periodontal diseases. Therefore, anti-inflammatory diet therapy contributes to the reduction of gingival inflammation and thus contributes to the reduction of the development of pathogenic bacteria in the gingival, responsible for the development of periodontal disease and directly by other chronic diseases.

## 1. Introduction

Idiopathic periodontitis is increasingly associated with pathogenic bacteria. The bacteria responsible for the development of periodontitis have reached the center of research in the last decade. They have been shown to be involved in the development of several chronic diseases, such as polycystic ovary or dyslipidemia [1]. These associated inflammatory diseases are of metabolic origin [2] Inflammation, not only gingival, is correlated with the presence of bacteria in the gingival crevicular fluid [3]. Some pathogens have been linked to Alzheimer’s (*Porphiromonas gingivalis*) [3,4] or cancer (*Fusobacterium nucleatum*, *Treponema denticola*) [5]. An important step in prevention was the possibility to test the composition of gingival crevicular fluid [6]. These analyzes test bacterial DNA, of species directly correlated with periodontal diseases, and indirectly correlated with increased levels of inflammatory mediators [7].

The classification and grading of periodontitis are important for the classification of periodontitis. According to this, periodontitis can be initial, moderate, or severe. Another classification is made according to the stages of periodontitis, where stage 1 loss of interdental clinical attachment at the site of the largest loss is 1–2 mm, in stage 2 it is 3–4 mm, in stage 3 and 4 is <5 mm. Numerous criteria have also been established for delimiting the correct classification, taking into account the systemic risks as inflammation. The most commonly used biomarker is gingival crevicular fluid [8].

A readjusted classification from 2018 shows the stages (1–4) and depending on the location (generalized, molar-incisor distribution), but also their degree of progression (slow rate of progression, moderate rate of progression, and rapid rate of progression) [9].

MS is moving towards Diabetes and its associated diseases [10]. It is based on a proinflammatory process, in which cytokines (TNFα, IL6) are correlated with the development of several chronic diseases [11]. Prevention only with diet therapy and physical activity is too little when infections with bacteria related to maintaining or increasing inflammation are also present. However, the importance of diet therapy and sports is indispensable [12], only in some situations it seems insufficient.

Recent studies that have looked at obesity and associated diseases have shown a direct relationship between visceral fat, non-alcoholic fatty liver disease, and diverticulosis. The relationship between the intestinal microbiome and the metabolic syndrome has been followed since 2013 [13], but dysbiosis and non-alcoholic fatty liver disease and diverticulosis appear in the most recent studies [14]. It seems that the disease of the fatty liver, visceral fat, and diverticulosis is the consequence of a marked dysbiosis, present in the metabolic syndrome [15,16]. Figure 1 shows the correlations between obesity, proinflammatory process, insulin resistance and diabetes.

This paper studies the correlation between the level of inflammation of the metabolic syndrome and the presence of pathogenic bacteria present in the periodontal gingival crevicular fluid, as well as the link between allopathic treatment and the presence of periodontopathogens at 3 and 6 months.

Among the paraclinical parameters that follow the evolution of the metabolic syndrome are glycemia fructosamine (for blood glucose monitoring) Total cholesterol, triglycerides (for monitoring dyslipidemia) uric acid, urea nitrogen (for renal and hepatic function) amylase (pancreatic function) for inflammation phosphate.

Total cholesterol (125–250 mg/dL), triglycerides (50–130 mg/dL) for tracking dyslipidemia.

Fuctozamine (205–285 µmol/L) reflects the glycosylation of non-enzymatic blood proteins and does not interfere with harvesting stress. The rate of formation of fructosamine is proportional to glucose levels, very useful for MS patients with glucose (60–100 mg/dL).

Uric acid (NAM 3 to 7.5 mg/dL women from 2.6 to 6 mg/dL), urea nitrogen (8–20 mg/dL) (for the renal and hepatic function) urea forming the main place in the liver, after which it is eliminated by glomerular filtration through the kidneys, this being an indicator for liver or kidney diseases.

Lactate dehydrogenase (230–460 IU/L), obtained from the cleavage of the protein directly to the intestine, is an indicator used in diet therapy, that has both absorption and intestinal proteolytic enzymes, and is used for tracking renal and hepatic function.

Amylase (27–83 IU/L) is an enzyme that catalyzes the hydrolytic degradation of starch from glycogen and oligo and polysaccharides. Values indicate changes in pancreatic disorder.

Alkaline phosphatase (men 53–128 IU/L, women 42–98 IU/L) is an enzyme with three forms of isoenzyme, hepatobiliary, bone, and intestinal. Elevated levels may indicate biliary diseases, bone, or intestinal tract.

The aim of the study is the reduction in the pro-inflammatory process in MS and its progress in preventing diabetes or other metabolic diseases.

This study aims to establish whether or not there is a relationship between initial, minor periodontopathy and metabolic syndrome.

The goal is to reduce the proinflammatory process in the metabolic syndrome thus improving the clinical parameters (BMI, fat mass, visceral fat, hydration status) and paraclinical (Glucose, Cholesterol, Triglycerides, Uric acid, Urea nitrogen, Alkaline phosphatase, Amylase, Fructosamine, Lactate), reducing the progression of periodontal diseases.

## 2. Materials and Methods

### 2.1. Body Analysis of Patients with MS

The clinical and paraclinical analyzes were performed in a private medical office of nutrition in Oradea, Echo laboratories, and the sampling and dental support were provided by Dr. Călinescu Delia, in a private medical office of dentistry.

#### 2.1.1. Anthropometric Tests

We conducted a cross-sectional study on MS patients that were enrolled in a cohort study, to obtain the clinical characteristics of MS patients and adapted the diet therapy for each patient. The patients in the study followed a personalized diet, based on a healthy diet (intake of macronutrients in the percentage of 45–55% carbohydrates, 25–35% protein, and 15–20% lipids, hypocaloric, with a reduction in caloric intake by 200 kcal). The personalization consisted of testing from the venous/capillary blood a specific allergic reaction of type 3 and 4 IgG, of a number of 90 foods, foods specific to the local area, and cuisine. Foods that had a specific IgG reaction were removed from the diet for 3 months, reintroduced only occasionally, until the end of the research period. The clinical evaluation was performed with the Tanita MC780MA bioelectric impedance body analyzer (Tokyo, Japan) [17], and the results were evaluated using GMON 3.4.1 medical software (Chemnitz, Germany). BIA-type body analyzers are devices accepted by the WPHNA (World Public Health Nutrition Association) and were used to determine body composition with high accuracy. The margin of error was 0.1 kg. We followed the evaluation of the affinity for diet therapy with the non-invasive medical device Cnoga MTX (Or-Akiva, Israel), which helped to follow the changes of the clinical parameters as a whole, checking the oxygen saturation, the blood pressure, and the blood pH. Patients were evaluated on an empty stomach in the morning.

The diagnosis of the metabolic syndrome was made following the HOMA index, mixed dyslipidemia, hypertension, visceral fat, fat mass, and the ratio between extracellular water and total water. The criterion for including patients in the current study was a diagnosis of metabolic syndrome of at least one year, and to present an initial periodontopathy. Moderate and severe periodontitis were the exclusion criteria. This study followed the evolution of fat mass, visceral fat, and the ratio between extracellular water and total body water (ECW/TBW), these having the most obvious changes following diet therapy, and they represent the highest risk for the unfavorable evolution of metabolic syndrome.

We followed the variations in the four independent groups, depending on sex, age, rural/urban environment, clinical parameters such as BAM, weight status, fat mass, visceral fat, hydration status, (ECW/TBW), sarcopenic index, phase angle, basal metabolism (BMR), and pH, followed by MS.

The completion of the evaluation sheets of the patients included in the research study took place at the beginning and the end of the study period by the participants.

#### 2.1.2. Tracking Metabolic Parameters

The total mass of the cells was directly proportional to the bioimpedance phase or phase angle and is measured by the bioelectric impedance body analyzer. Patients were evaluated at the beginning and the end of the treatment with this device. With bare feet, they climbed on the base of the scale and held two electrodes in their hands. The measurements lasted up to 10 s after which the results were evaluated. According to the results, we could see their improvement or deterioration, and when analyzed in the statistical program SPSS 22 (New York, NY, USA), we could follow both the final result compared to the initial values and the possible correlations between health (measured by phase angle) and the healing of infections in patients with MS.

#### 2.1.3. Paraclinical Analyses

Automated analyzer for clinical chemistry-SPOTCHEM EZ SP-4430-ARKRAY Inc. Koka-Shi, Japan. Whole blood samples can be measured easily and promptly with a built-in centrifuge for pretreatment. Continuous measurement, up to 9 items, is available.

By reading the “magnetic card” attached to each test strip, variations among lots and changes with the timing of test strips are automatically calibrated. The users are totally free from complicated operations.

Improvement of the sampling mechanism almost doubled the measurement speed at its maximum compared to the current model.

Measurement principle: Optical measurement of reflection intensity-five different types of optical filters (five wavelengths), and the optimal wavelength is selected for each parameter measured (tested).

240 microliters of whole blood are collected. It is placed in a special tank with heparin and placed in the device. The analyzer has a built-in centrifuge, processes the sample, and in 10 min the result is printed by the built-in printer.

Reagent type: dry chemistry

### 2.2. Micro-IDent Test

A DNA test of bacteria present in the gingival crevicular fluid.

Gingival fluid is collected using dry paper dots, when pocket depth > 4 mm with BOP (despite excellent oral hygiene). Dry paper dot is held in pocket depth for 15 s until it completely absorbs gingival crevicular fluid. The paper dot is placed in the kit box, closed, and stored between 2–8° Celsius until processing.

Extraction and handling of DNA from dry paper dots.

A bacterial DNA extraction kit (HAIN Lifescience) from dry paper dots was used for extraction according to the manufacturer’s instructions. Briefly, 100 μL of lysis solution was added to the flask containing each periodontal sample and vigorously for 10 s to elute the bacterial cells from the paper dots. It was incubated at 95 °C on the PeQLab heating block (Biotechnology, GmbH, Germany) for 5 min with a strong vortex every 2 min. The mixture was then cooled to room temperature.

Then added a volume of neutralizing buffer of 100 μL and then spun into microfuge Mikro 200 (Centrifugen, Hettich, Germany) for 10 min at 14,000 rpm. A volume of 5.0 μL of supernatant was used as a template for the subsequent amplification step. It was stored at −20 °C pending hybridization.

16-rDNA Micro-IDent^®^plus-Multiplex PCR.

PCR amplification was performed in a 50 μL reaction volume consisting of 5.0 μL of template DNA and 45 μL of reaction mixture containing 35 μL of primer-nucleotide-PNM (Micro-IDent^®^plus) mixture, 5.0 μL of 10× PCR buffer, 5.0 μL of 2.5 mM MgCl2 and 0.2 U Taq at the hot start (Qiagen, GmbH, Germany). PCR cycling was performed in GTQ-Cycler 96 thermal cycler (HAIN Lifescience, Germany). Cycling conditions included an initial stage of denaturation at 95 °C for 5 min, 10 cycles at 95 °C for 30 sec and at 60 °C for 2 min, 20 cycles at 95 °C for 10 s, at 55 °C for 30 s and at 72 °C for 30 s and a final step extension to 72 °C for 10 min. Negative control was included in the test sample. Negative control was 5.0 μL of sterile water PCR, each added to 45 μL of the reaction mixture. Subsequently, the reverse hybridization was performed according to the Micro-IDent^®^plus test (HAIN Lifescience GmbH, Nehren, Germany). Reverse hybridization Subsequent reverse hybridization was performed according to the Micro-IDent^®^plus. Briefly, amplicon biotinylate was denatured and incubated in Twincubator^®^ (HAIN Lifescience) at 45 °C with hybridization buffer and coated strips with two control lines (Conjugates and amplification) with six (Micro-IDent^®^plus) species-specific probes.

After the PCR products were bound to their respective probe complements, a very specific washing step removed any nonspecific bound DNA. Alkaline phosphatase-conjugated to streptavidin was added, the samples were washed and the hybridization products were visualized by adding an alkaline phosphatase substrate. A total of 11 selected species of periodontopathogen bacteria can be identified using the test [18].

## 3. Results

The study involved 111 people with MS, divided into 3 groups according to preference and stage of the disease.

Control group (CG) of 36 people

Diet therapy group (DG) 35 people

Diet therapy and sports group (DSG) of 40 people

The physical activity (sport) followed was light, for cardiovascular stimulation, for 30–60 min, 2–3 times a week, in DSG.

CG did not follow any diet therapy or regular physical activity.

From the cohort, 36 people performed the periodontopathogens test (Micro-IDent), chosen injured.

The initial dates shows that the cohort consists of 87 men and 24 women (*t* = 30.986, *p* = 0.001), of which 48 from urban and 63 from rural areas (*t* = 33.186, *p* = 0.001), with an average age of 39.45 (SD 15.01) years (*t* = 27.687, *p* = 0.001). The initial anthropometric parameters were thus the mean BMI on cohort 30.58 (SD 7.67) *t* = 41.953 *p* = 0.001, fat mass 30.79 (SD 8.78) *t* = 36.948 *p* = 0.001, visceral fat 7.56 (SD 5.43) *t* = 14.674 *p* = 0.001, and ECW/TBW 42.86 (SD 3.17) *t* = 142.190 *p* = 0.001. The statistical description of the initial data for each group is presented in Table 1 and Table 2.

### 3.1. Evolution of Parameters

#### 3.1.1. Results (Evolution) of Anthropometric Analyses

The results obtained at the end of the research period compared to the initial values are statistically processed, and presented in Figure 2 by presentation with lines.

The graphical representation of the mean values of BMI, fat mass, visceral fat, and ECW/TBW, described in Table 3 looks like this: in CG, a statistically significant worsening of the parameters can be observed (*p* < 0.05) which can be seen from the negative value of “*t*” in the case of BMI, fat mass and visceral fat.

In the case of DG, anthropometric parameters such as BMI, fat mass, visceral fat, and ECW/TBW improved statistically significantly (*p* < 0.05).

The DSG group can see a more pronounced improvement, statistically significant (*p* < 0.05), the best result is obtained in this category except for BMI, which has better values in the DG group. The text continues here.

Regarding BMI, we used the ANOVA statistical test for 2 independent groups and we obtained F = 15,448, *p* = 0.001, so in the post-test (after 6 months) BMI differs significantly in the 2 groups, as can be seen in Table 3. Using BONFERRONI Post Hoc tests, the data obtained indicate significant differences between groups as follows: between CG and DG *p* = 0.001, but also between CG and DSG *p* = 0.001. Insignificant differences were found between DG and DSG groups *p* > 0.05.

Following the fat mass, we used the ANOVA statistical test for 2 independent groups and we obtained F = 23.836, *p* = 0.001, so in the post-test (after 6 months) the fat mass differs significantly in the 2 groups, as can be seen in Table 3. Using BONFERRONI Post Hoc tests, the data obtained indicate significant differences between groups as follows: between CG and DG *p* = 0.018, but also between CG and DSG *p* = 0.001. Significant differences were also registered between the DG and DSG groups *p* = 0.001.

Checking the visceral fat using the ANOVA test for 2 independent groups and we obtained F = 12.105, *p* = 0.001, so in the post-test (after 6 months) the visceral fat differs significantly in the 2 groups, as can be seen in Table 3 With the help of BONFERRONI Post Hoc tests, the data obtained indicate insignificant differences between CG and DG *p* = 0.905, but between CG and DSG *p* = 0.001 changed significantly. Significant differences were also recorded between DG and DSG groups *p* = 0.002.

In terms of the ratio between ECW/TBW, using the ANOVA test for 2 independent groups and we obtained F = 5.051, *p* = 0.005, so in the post-test (after 6 months) ECW/TBW differs significantly in the 2 groups, after as can be seen in Table 3. Using BONFERRONI Post Hoc tests, the data obtained indicate insignificant differences between CG and DG *p* = 0.533, but between CG and DSG *p* = 0.006 changed significantly. Insignificant differences were also registered between DG and DSG groups *p* = 0.249.

#### 3.1.2. The Result of Paraclinical Analyzes

Glucose, Cholesterol, Triglycerides, Uric acid, Urea nitrogen, Alkaline phosphatase, Amylase, Fructosamine, and Lactate dehydrogenase were monitored within the paraclinical parameters.

Figure 2 shows a significant decrease in the values of paraclinical parameters at the end of the study in DG and DSG, but also a stagnation or even aggravation of the results in GC.

Thus in CG in the case of glucose and lactate dehydrogenase no change was observed at the end of the research period, cholesterol by 7 units, uric acid by 1.2 units, and alkaline phosphatase by 10 units decreased but not reaching the thresholds of significance, and a slight increase in triglycerides with 5 units, urea nitrogen with one unit, amylase with 3 and fructosamine with 4 units, but still statistically insignificant.

Following DG, the following parameters decrease blood glucose by 15 units, cholesterol 154 units, triglycerine 72 units, uric acid 4.8 units, alp 42 units, fructosamine 61 units, dehydrated lactate 192 units, and urea nitrogen increased by 8 units, and amylase increased by 19 units.

At DSG parameters decreased significantly as follows: Glucose with 15 units, Cholesterol 145 units, Triglycerides 83 units, Uric acid 3.1 units, Alkaline phosphatase 53 units, Fructosamine with 60 units, Lactate dehydrogenase 177 units. Urea nitrogen increased by 12 units, but Amylase also increased by 20 units.

#### 3.1.3. Micro-IDent Tests Results

At the end of the study, the presence of pathogenic bacteria in the periodontal gingival crevicular fluid changed statistically significantly in the cohort (*p* = 0.001). From the point of view of the groups, in CG it did not change significantly (*p* = 0.324), in DG the presence of infection decreased significantly at the end of the study (*p* = 0.230), and in DSG the infection was also statistically significant (*p* = 0.044). In CG the infection was reduced by 0.9%, in DG 4.51%, in DSG by 3.6%, shown graphically in Figure 3.

Following the statistical processing, presented in Figure 4 and Figure 5 people had infection with 3 species of pathogens, in 12 people it tested positive for 6 pathogens, 6 people had infections with 7 bacteria, another 6 people with 8 bacteria, and 11 types of pathogenic bacteria were present in 6 people. One person was infected with several bacteria. Thus, at the end of the intervention it was reduced by 9.01% of the total cohort, but by 27.77% compared to the total number of infected, described by groups in Table 4.

The results of this power analysis underline that a satisfactory degree of power was obtained by increasing the size of the batch, the result being 0.805 i.e., 80.5% for periodontal diseases at the end of the research period.

In the case of bacteria, *Eubacterium nodatum* was present in 6 people, *Eikenella corrodens* and *Prevotella intermedia* in 12 people each, *Capnocytophaga* spp. and *Porphyromonas gingivalis* in 18 people, *Peptostreptococcus micros* and *Campylobacter rectus* in 24 people, *Actinobacillus actinomycetemides*, *Bacteroides forsythus*, and *Treponema denticola* in 30 people, and Fusobacterium nucleatum in 36 people was detected. The description of the number of patients for each pathogen divides can be verified in Figure 6, and the percentage distribution of patients with infection with each pathogen divides in Figure 7.

#### 3.1.4. Pearson Correlation

The correlation between the evolution of periodontopathogens and the evolution of fat mass, visceral fat, and ECW/TBW ratio was followed. Table 5 processed the Pearson correlation on the cohort, and a negative relationship was obtained. As fat mass decreases, visceral fat and ECW/TBW ratio also decrease periodontopathy, but without statistical significance.

The Pearson correlation regarding the evolution of periodontopathogens correlated with the evolution of fat mass, visceral fat, and ECW/TBW ratio are presented in Table 6.

Thus in CG, the relationship between periodontopathogens and ECW/TBW shows a strongly negative relationship r = −0.452, statistically significant (*p* < 0.05), presented in Figure 7. The lower the ECW/TBW ratio, which indicates inflammation, increases periodontopathy to this group. The first quarter in the category of infection present in CG at the bottom indicates an inversely proportional relationship. Fat mass and visceral fat have a directly proportional relationship with periodontopathogens, but without reaching the thresholds of statistical significance. So as fat mass and visceral fat increase, periodontal disease also increases.

In the case of DG and DSG, none of the parameters reached the significance thresholds, but show a relationship directly proportional to fat mass and visceral fat, and inversely proportional to the ratio between ECW/TBW.

## 4. Discussion

MS is one of the most common diseases in the last decade [19]. Cardiovascular risk is the main complication, in addition to obesity, hypertension, or dyslipidemia [20]. Nazare in a 2014 study looked at BMI, the mass found visceral fat to assess cardiovascular risk [21]. In the current study, the evolution of the metabolic syndrome is followed by clinical parameters, such as BMI, fat mass, visceral fat or ECW/TBW [22]. The evolution of BMI following the dietary intervention was followed in numerous studies [23,24,25], in which it was observed a negative evolution of it has beneficial effects either in the case of diabetes or in people with insulin resistance. BMI in our research study decreased significantly following diet therapy but to a greater extent associated with sports. This result reflects that clinical diet therapy with regular sports can bring greater benefits than just diet therapy.

The microbiome has a particularly important role in the evolution of metabolic syndrome. Recent studies show a link between metabolic syndrome and inflammation and bacteria in reducing intestinal inflammation [26,27]. This approach should not be missing from the management of the metabolic syndrome, especially where the provision of dietary fiber, with a role in the development and feeding of the microbiome, is insufficient. Our study, without the additional intake of pre-/probiotics, can be continued with the recommendation of probiotics for amplification, being able to highlight the role of the microbiome.

To improve the results, recent studies suggest a possible association between *Konjac glucomannan* (butyrate producer), galactooligosaccharides (GOS), fructooligosaccharides (FOS), zinc citrate, cholecalciferol (vitamin D3), riboflavin (vitamin B2) [28,29,30]. Among the bacteria *Akkermansia muciniphila* and *Faecalibacterium prausnitzii* may bring additional benefits [26,31]. For periodontal diseases we can obtain additional benefits with herbal infusion in oral health care [32].

The importance of sport in the management of metabolic syndrome was also emphasized by Schmidt-Trucksäss in 2006 [33], and in 2016 Bird studied the insulin sensitivity at the muscle level observed lower in those who performed regular physical activity [34]. Beneficial effects of physical activity are highlighted by several studies [35,36], but the present study also obtained superior results in the DSG group in the case of most clinical parameters, but also in the case of paraclinical parameters.

A study published by Oxford Academic in 2013, highlighted the clear link between MS and periodontitis [37]. High levels of cytokines and chemokines have also been observed in patients with periodontitis in several studies [38,39]. The link between cytokines and fat mass is obvious [40], which is linked in our study between fat mass and periodontitis. The inflammation resulting from the ratio between ECW/TBW has been studied since 2017 [41,42], used with great pleasure for stable and correct results of bioelectric impedance devices. In our study, a statistically significant link was observed between water retention correlated with inflammation and the presence of periodontopathogens in patients with MS.

Visceral fat is directly related to obesity, but also to the risk of developing diabetes [43,44]. High levels of visceral fat in MS patients are also correlated with high levels of cytokines [45].

A clinical diet helps reduce periodontitis [46]. The general reduction of inflammation by reducing BMI, visceral fat mass leads to a reduction in the HOMA index, an indicator of metabolic syndrome [36,47]. 

The link between glycosylated hemoglobin and periodontal disease was followed in a 2021 study, and a significant decrease in glycosylated hemoglobin was found in the group with rigorous oral hygiene, compared to the control group [48]. In the current study, we can follow the improvement of paraclinical parameters (Glucose, Cholesterol, Triglycerides, Uric acid, Urea nitrogen, Alkaline phosphatase, Amylase, Fructosamine, Lactate dehydrogenase).

Bacterial proteins of *Bacteroides forsythus* bind to Toll-like receptors to induce proinflammatory cytokines, thus correlating its presence with high levels of cytokines [49]. This bacterium was present in 30 patients at the beginning of the study.

Another pathogen is correlated with a high level of cytokines. It is *Campylobacter rectus* that has been followed due to gingival inflammation [50], and the presence of *Porphiromonas gingivalis* has opened new studies regarding inflammation [51]. In the current study, *Campylobacter rectus* was present in 21.62% of the cohort, while *Porphiromonas* was present in 16.21% of the cohort.

The adjusted R tells us how much variance in the outcome would be accounted for if the model has been derived from the population from which the sample was taken. A 2010 study looked at the risk of developing metabolic syndrome in patients with schizophrenia, tracking genotype, and statistically processed with power analysis [52]. The results thus received could be extrapolated to the population from which the individuals were extracted, as in our case, where the probability of developing periodontal diseases in people with metabolic syndrome was followed. The results showed a rate of 80.5% similar to the cohort studied.

A study published in 2021, followed menopausal women in Korea, verifying the link between periodontal disease and metabolic syndrome [53]. The association between the 2 parameters proved, but in the case of the Koreans, it did not reach the significance thresholds. It can be explained that the prevalence of obesity was 33.2%. In Europe, the prevalence of obesity is 40–59.9%. Following only the adult population, due to sedentary lifestyle, this percentage is over 60%, according to WHO estimates [54]. Thus, comparing the insignificant relationship of metabolic syndrome and periodontitis in Koreans, and the significant relationship obtained in adults in Romania, the key is the much higher rate of obesity, so it can be seen that obesity-related diseases are amplified.

The evolution of periodontal diseases, as well as the evolution of metabolic syndrome, is obvious in case it is not intervened [36,55]. This advancement was accentuated in CG, who did not perform any allopathic or dietary treatment. This is conclusive from the negative Pearson correlation value. In the 2 research groups, no significant differences in the evolution of infections with pathogens in the gingival crevicular fluid were observed.

## 5. Conclusions

Each parameter followed was statistically significantly modified (*p* < 0.05) for each batch. For CG, the changes show that without any intervention MS is advancing, the results being worsened.

Paraclinical parameters improved as in DG and DSG, however in CG they did not change statistically significantly (*p* > 0.05). The best results were obtained at DSG, with 2.6% better than at DG, which shows the importance of physical activity.

At DG, the best results were obtained in terms of reducing periodontal diseases. The more weighted decrease in DSG can be correlated with the presence of physical activity, the increase in muscle mass, which implies a higher level of inflammation (for example muscle fever).

There was a statistically significant worsening of periodontopathogens correlated with the development of metabolic syndrome (increase in fat mass, visceral fat, and the ratio between ECW/TBW) in the CG group, and in the case of DG and DSG groups, there was an improvement in parameters including periodontitis. 

With applied diet therapy combined with sports, in the case of initial periodontopathy in patients with MS, the phase has also improved, indicating an increase in health and an improvement in cellular functions. However, additional studies, longer study periods, and a larger number of patients are needed to determine the exact mechanisms underlying these correlations.

## 6. Patents

This section is not mandatory but may be added if patents are resulting from the work reported in this manuscript.

## Figures and Tables

**Figure 1 biomedicines-09-01709-f001:**
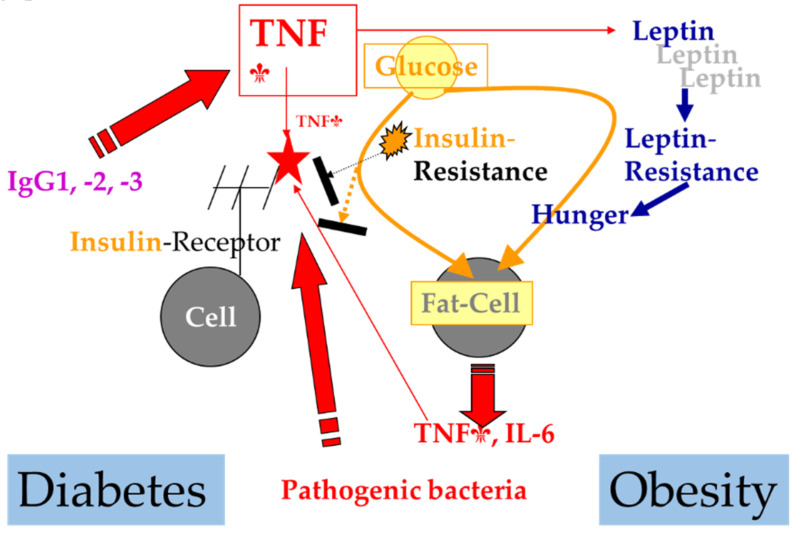
The evolution of the metabolic syndrome in the presence of pathogenic bacteria in the gingival crevicular fluid. The red arrow points a proinflammatory process, which leads to the secretion of cytokines (red star). The yellow arrow shows the relationship between fat mass and insulin resistance, and the blue arrow shows hormonal changes (leptin) caused by cytokines, which leads to overeating, and thus increased fat mass.

**Figure 2 biomedicines-09-01709-f002:**
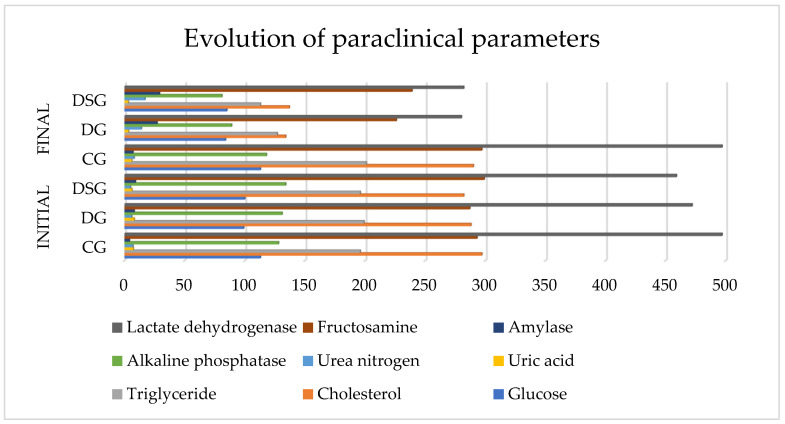
Evolution of paraclinical parameters in all three research groups.

**Figure 3 biomedicines-09-01709-f003:**
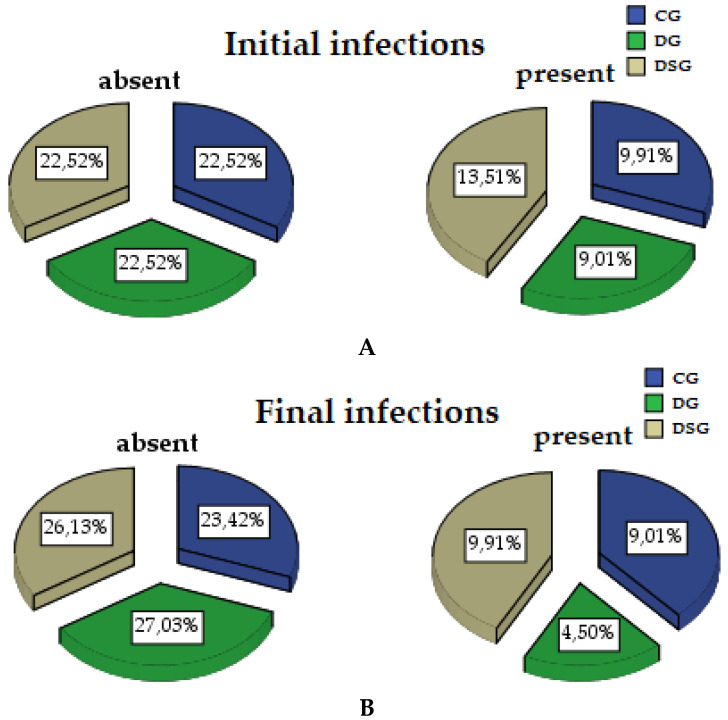
The evolution of the infection of the gum to the three research groups com-sounding initial infection (**A**) with the final (**B**).

**Figure 4 biomedicines-09-01709-f004:**
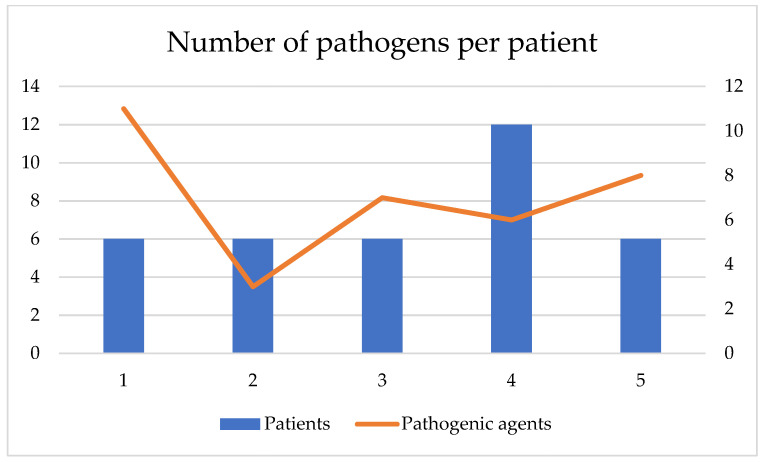
Graphic presentation of the correlation of patients with the number of pathogens per patient.

**Figure 5 biomedicines-09-01709-f005:**
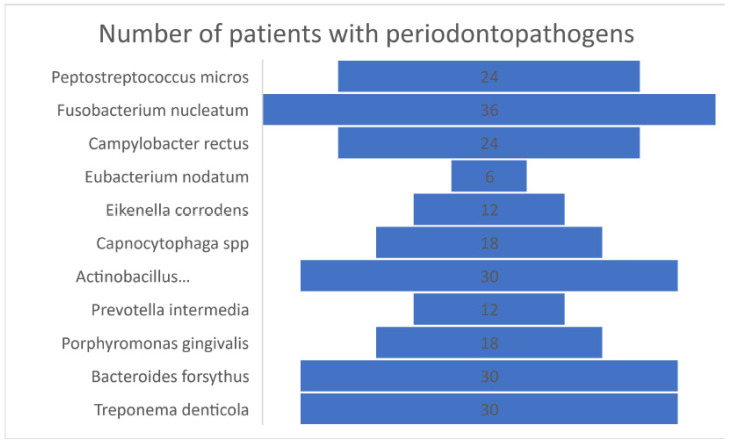
Graphic description of the number of patients in the study with the presence of pathogens.

**Figure 6 biomedicines-09-01709-f006:**
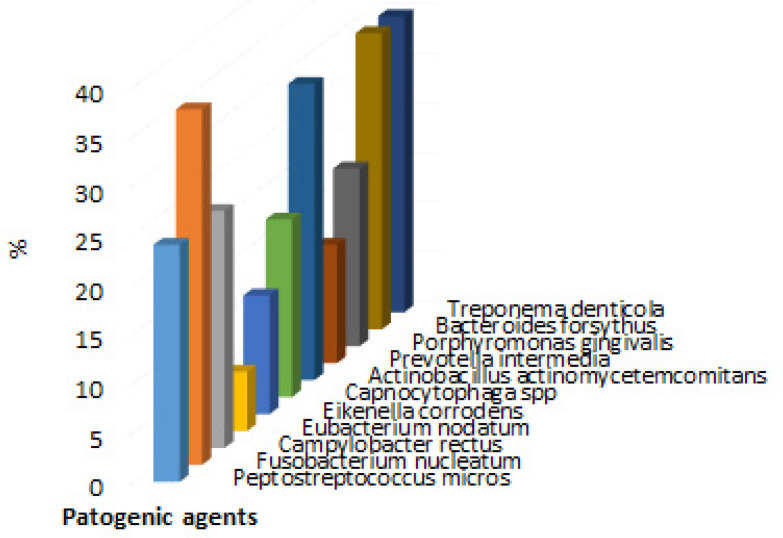
Graphic description of the number of patients in percentages who presented infections with pathogens.

**Figure 7 biomedicines-09-01709-f007:**
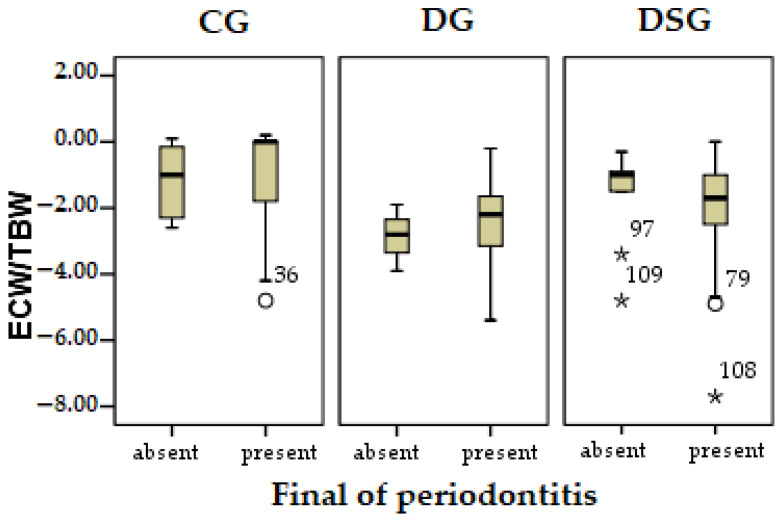
Evolution of the correlation of fat mass with periodontopathogens at the end of the study in the three research groups.

**Table 1 biomedicines-09-01709-t001:** Initial parameters about gender, source environment, and periodontopathogens.

Initial Parameters		Groups					
		CG		DG		DSG	
		N	%	N	%	N	%
Gender	Men	28	77.77	29	82.86	30	75.00
	Women	8	22.22	6	17.14	10	25.00
Area of provenience	Urban	12	33.33	14	40.00	22	55.00
	Rural	24	66.66	21	60.00	18	45.00
Periodontopathogens	Absent	25	69.44	25	71.43	25	62.50
	Present	11	30.55	10	28.57	15	37.50

CG = Control group; DG = Diet therapy group; DSG = Diet therapy and sport group; N = Number of patients.

**Table 2 biomedicines-09-01709-t002:** Initial age and clinical parameters.

Initial Parameters	Groups								
	CG			DG			DSG		
	N	Mean	SD	N	Mean	SD	N	Mean	SD
Age	36	42.67	15.64	35	42.31	17.95	40	34.05	9.40
Initial BMI	36	29.29	5.60	35	30.95	6.50	40	31.42	9.95
Initial fat mass	36	32.27	7.82	35	33.71	7.10	40	26.92	9.67
Initial visceral fat	36	8.78	4.87	35	9.37	6.70	40	4.90	3.36
Initial ECW/TBW	36	43.21	2.83	35	43.69	3.13	40	41.85	3.31

CG = Control group; DG = Diet therapy group; DSG = Diet therapy and sport group; N = Number of patients; SD = Standard deviation.

**Table 3 biomedicines-09-01709-t003:** Statistical evaluation of research parameters compared final results with initial values in all three research groups.

Groups			Mean	N	Std. Deviation	Correlation	*t*	Sig.
CG	Pair 1	Initial BMI	29.2889	36	5.59682	0.999 **	−3.897	0.001
		Final BMI	29.4667	36	5.79300			
	Pair 2	Initial fat mass	32.2667	36	7.82407	1.000 **	−4.799	0.001
		Final fat mass	32.3222	36	7.82023			
	Pair 3	Initial visceral fat	8.7778	36	4.87038	0.999 **	−2.092	0.044
		Final visceral fat	8.8889	36	5.11456			
	Pair 4	Initial ECW/TBW	43.2056	36	2.82539	0.865 **	3.741	0.001
		Final ECW/TBW	42.3000	36	2.76106			
DG	Pair 1	Initial BMI	30.9497	35	6.49928	0.862 **	8.903	0.001
		Final BMI	24.7031	35	3.08141			
	Pair 2	Initial fat mass	33.7057	35	7.10435	0.719 **	7.118	0.001
		Final fat mass	27.7371	35	5.58633			
	Pair 3	Initial visceral fat	9.3714	35	6.69981	0.989 **	7.242	0.001
		Final visceral fat	7.7429	35	5.73607			
	Pair 4	Initial ECW/TBW	43.6886	35	3.13320	0.941 **	12.180	0.001
		Final ECW/TBW	41.2686	35	3.45805			
DSG	Pair 1	Initial BMI	31.4205	40	9.95106	0.840 **	5.636	0.001
		Final BMI	24.7725	40	3.21248			
	Pair 2	Initial fat mass	26.9160	40	9.66844	0.835 **	6.315	0.001
		Final fat mass	21.4950	40	6.97593			
	Pair 3	Initial visceral fat	4.9000	40	3.35735	0.897 **	4.210	0.001
		Final visceral fat	3.9000	40	2.79009			
	Pair 4	Initial ECW/TBW	41.8475	40	3.31012	0.894 **	7.758	0.001
		Final ECW/TBW	39.9725	40	3.33459			

N = Number of patients; *t* = Coefficient for the T-Student statistical test; Sig. = Statistic signification; ** = Correlation is significant at the 0.01 level (2-tailed); CG = Control group; DG = Diet therapy group; DSG = Diet therapy and sport group.

**Table 4 biomedicines-09-01709-t004:** The final results of periodontal diseases in the three groups at the end of the research period.

Periodontopathogens		Final Periodontopathogens	
		Absent	Present
		N	N
Groups	CG	26	10
	DG	30	5
	DSG	29	11

N = Number of patients; CG = Control group; DG = Diet therapy group; DSG = Diet therapy and sport group.

**Table 5 biomedicines-09-01709-t005:** Pearson correlation between periodontal disease and ECW/TBW fat mass in the cohort.

Pearson Correlation		Periodontopathogens
BMI	r	0.102
	Sig.	0.286
	N	111
Fat mass difference	r	−0.023
	Sig.	0.813
	N	111
Visceral fat difference	r	−0.133
	Sig.	0.164
	N	111
ECW/TBW difference	r	−0.069
	Sig.	0.471
	N	111

r = Pearson Coefficient; N = Number of patients; Sig. = Statistic signification.

**Table 6 biomedicines-09-01709-t006:** Pearson correlation between the evolution of periodontitis and grease mass and ECW/TBW in all three research groups.

Groups			Periodontopathogens
CG	Fat mass difference	r	0.166
		Sig.	0.332
	Visceral fat difference	r	0.273
		Sig.	0.107
	ECW/TBW difference	r	−0.452 **
		Sig.	0.006
	Patients	N	36
DG	Fat mass difference	r	0.062
		Sig.	0.724
	Visceral fat difference	r	0.009
		Sig.	0.960
	ECW/TBW difference	r	−0.197
		Sig.	0.258
	Patients	N	35
DSG	Fat mass difference	r	0.076
		Sig.	0.640
	Visceral fat difference	r	0.156
		Sig.	0.337
	ECW/TBW difference	r	−0.056
		Sig.	0.733
	Patients	N	40

** Correlation is significant at the 0.01 level (2-tailed); r = Pearson Coefficient; N = Number of patients; Sig. = Statistic signification; CG = Control group; DG = Diet therapy group; DSG = Diet therapy and sport group.

## Data Availability

Data supporting the reported results can be found in the archives ECHO LABORATORIES, for the years 2019–2021.

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
