# Peer review of "Correlation of Periodontal Bacteria with Chronic Inflammation Present in Patients with Metabolic Syndrome"

_biomedicines, 2021, doi:10.3390/biomedicines9111709_

Round 1
Reviewer 1 Report
This is a study that sought to examine the effect of differential treatment (diet, exercise etc)
on the parameters of metabolic syndrome and associated clinical and periopathogenic data.
The manuscript in itself encompasses a substantial dataset and obviously reflects the hard
work and effort put in by the investigators. However, overall, I find this manuscript to be
haphazard, lacking detail and poorly written. Thus, I do not think this manuscript is fit for
publication as it stands now.
Some of the main concerns I have are the following:
Inadequate description of groups and how they were recruited.
No periodontal assessment or dental data
Lack of sample size calculation
Poor statistical description
Unclear aims and objectives
Poor description of methodology- eg how crevicular fluid was collected
Conclusion is reiteration of results and the final sentence does not link with the current
study
References are incomplete and inconsistence
Author Response
Response to Reviewer 1
Firstly, I, the author of the present manuscript wish to thank you for thoughtful commentary you have provided to improve the quality of the paper. We are very grateful for the time and effort you have devoted to this task. We have extensively revised our manuscript according to the recommendations. All changes in the text and the new figures that we have redesigned are highlighted. Please, see the point-by-point answers to your comments below. All correction was highlighted in the manuscript.
Comment 1.: Inadequate description of groups and how they were recruited.
Answer 1.:
Thank you for the remark. The manuscript was completed with the detailed description of the research groups, and with the inclusion / exclusion criteria. (lines 117-124, 203-205).
„The diagnosis of the metabolic syndrome was made following the HOMA index, mixed dyslipidemia, hypertension, visceral fat, fat mass, but also the ratio between extracellular water and total water. The criterion for including patients in the current study was a diagnosis of metabolic syndrome of at least one year, and to present an initial periodontopathy. Moderate and severe periodontitis were the exclusion criteria. This study followed the evolution of fat mass, visceral fat, and the ratio between ECW / TBW, these having the most obvious changes following diet therapy, and they represent the highest risk for the unfavorable evolution of metabolic syndrome.”
„The physical activity (sport) followed was light, for cardiovascular stimulation, for 30-60 minutes, 2-3 times a week, in DSG.
CG did not follow any diet therapy or regular physical activity.”
Comment 2.: No periodontal assessment or dental data
Answer 2.:
Thank you for the amendment. I complete the manuscript with the required information. Please, see the correction highlighted in the manuscript (lines 36-45, 119-122).
„The classification and grading of periodontitis is important for the classification of periodontitis. According to this, periodontitis can be initial, moderate or severe. Another classification is made according to the stages of periodontitis, where stage 1 loss of inter-dental clinical attachment at the site of the largest loss is 1-2 mm, in stage 2 it is 3-4 mm, in stage 3 and 4 is <5 mm . Numerous criteria have also been established for delimiting the correct classification, taking into account the systemic risks as inflammation. The most commonly used biomarker is gingival crevicular fluid.
A readjusted classification from 2018 shows the stages (1-4) and depending on the location (generalized, molar-incisor distribution), but also their degree of progression (slow rate of progression, moderate rate of progression, and rapid rate of progression).”
„The criterion for including patients in the current study was a diagnosis of metabolic syn-drome of at least one year, and to present an initial periodontopathy. Moderate and severe periodontitis were the exclusion criteria.”
Comment 3.:
Lack of sample size calculation.
Answer 3.: Thank you for the suggestion. I complete the manuscript with the One-sample test analysis (lines 208-214). Please, see the correction highlighted in the manuscript.
„The initial dates shows that the cohort consists of 87 men and 24 women (t= 30,986, p=0.001), of which 48 from urban and 63 from rural areas (t= 33,186, p=0.001), with an average age of 39.45 (SD 15.01) years (t=27.687, p=0.001). The initial anthropometric pa-rameters were thus the mean BMI on cohort 30.58 (SD 7.67) t=41.953 p=0.001, fat mass 30.79 (SD 8.78) t=36.948 p=0.001, visceral fat 7.56 (SD 5.43) t=14.674 p=0.001, and ECW/TBW 42.86 (SD 3.17) t=142.190 p=0.001. The statistical description of the initial data for each group is presented in tables 1 and 2.”
Comment 4.: Poor statistical description
Answer 4.:
Thank you very much for the amendment. I complete the manuscript with the test statistic ANOVA, BONFERRONI, and POWER analysis (lines 228-251, 285-287).
„Regarding BMI, we used the ANOVA statistical test for 2 independent groups and we obtained F = 15,448, p = 0.001, so in the post-test (after 6 months) BMI differs significantly in the 2 groups, as can be seen in table 3. Using BONFERRONI Post Hoc tests, the data obtained indicate significant differences between groups as follows: between CG and DG p = 0.001, but also between CG and DSG p = 0.001. Insignificant differences were found between DG and DSG groups p> 0.05.
Following the fat mass, we used the ANOVA statistical test for 2 independent groups and we obtained F = 23.836, p = 0.001, so in the post-test (after 6 months) the fat mass differs significantly in the 2 groups, as can be seen in table 3. Using BONFERRONI Post Hoc tests, the data obtained indicate significant differences between groups as follows: between CG and DG p = 0.018, but also between CG and DSG p = 0.001. Significant differences were also registered between the DG and DSG groups p = 0.001.
Checking the visceral fat using the ANOVA test for 2 independent groups and we obtained F = 12.105, p = 0.001, so in the post-test (after 6 months) the visceral fat differs significantly in the 2 groups, as can be seen in Table 3 With the help of BONFERRONI Post Hoc tests, the data obtained indicate insignificant differences between CG and DG p = 0.905, but between CG and DSG p = 0.001 changed significantly. Significant differences were also recorded between DG and DSG groups p = 0.002.
In terms of the ratio between ECW / TBW, using the ANOVA test for 2 independent groups and we obtained F = 5.051, p = 0.005, so in the post-test (after 6 months) ECW / TBW differs significantly in the 2 groups, after as can be seen in Table 3. Using BONFERRONI Post Hoc tests, the data obtained indicate insignificant differences between CG and DG p = 0.533, but between CG and DSG p = 0.006 changed significantly. Insignificant differences were also registered between DG and DSG groups p = 0.249.”
„The results of this power analysis underline that a satisfactory degree of power was obtained by increasing the size of the batch, the result being 0.805 ie 80.5% for periodontal diseases at the end of the research period.”
Comment 5.: Unclear aims and objectives
Answer 5.:
Thank you very much for the suggestion, the obiectives were modified accordingly (lines 85-92).
„The aim of this study is to establish whether or not there is a relationship between initial, minor periodontopathy and metabolic syndrome.
The goal is to reduce the proinflammatory process in the metabolic syndrome thus improving the clinical parameters (BMI, fat mass, visceral fat, ECW / TBW) and paraclinical (Glucose, Cholesterol, Triglycerides, Uric acid, Urea nitrogen, Alkaline phosphatase, Amylase, Fructosamine, Lactate) , reducing the progression of periodontal diseases.”
Comment 6.: Poor description of methodology- eg how crevicular fluid was collected
Answer 6.:
Gratefully accepting the observation, describe in detail the collection of crevicular fluid (lines 161-164)
„Gingival fluid is collected using a dry paper dots, when pocket depth> 4mm with BOP (despite excellent oral hygiene). Dry paper dot is held in pocket depth for 15 seconds, until it completely absorbs gingival crevicular fluid. The paper dot is placed in the kit box, closed, and stored between 2-8° Celsius until processing.”
Comment 7.: Conclusion is reiteration of results and the final sentence does not link with the current study
Answer 7.:
Again, we agree with your suggestion, and correction was made accordingly (lines 474-478).
„With applied diet therapy combined with sports, in the case of minor periodontitis in patients with MS, the phase has also improved, indicating an increase in health and an improvement in cellular functions. However, additional studies, longer study periods, and a larger number of patients are needed to determine the exact mechanisms underlying these correlations.”
Comment 8.: References are incomplete and inconsistence
Answer 8.:
Thank you very much for the comment. I completed the manuscript with another 17 references (8,9,13,14,15,16,26,27,28,29,30,31,32,48,52,53,54).
Reviewer 2 Report
Thank you so much for letting me review this interesting work
Abstracts and keywords spelled correctly
To be included in the introduction the new classification of periodontal disease where they compare the various systemic diseases,
Response to Reviewer 2
Firstly, I, the author of the present manuscript wish to thank you for thoughtful commentary you have provided to improve the quality of the paper. We are very grateful for the time and effort you have devoted to this task. We have extensively revised our manuscript according to the recommendations. All changes in the text and the new figures that we have redesigned are highlighted. Please, see the point-by-point answers to your comments below. All correction was highlighted in the manuscript.
Comment 1.: I am attaching references: Tonetti MS, Greenwell H, Kornman KS. Staging and grading of periodontitis: Framework and proposal of a new classification and case definition. J Periodontol. 2018 Jun; 89 Suppl 1: S159-S172. Caton JG, Armitage G, Berglundh T, Chapple ILC, Jepsen S, Kornman KS, Mealey BL, Papapanou PN, Sanz M, Tonetti MS. A new classification scheme for periodontal and peri-implant diseases and conditions - Introduction and key changes from the 1999 classification. J Periodontol. 2018 Jun; 89 Suppl 1: S1-S8.
Answer 1.:
Thank you very much for the suggestion. I will add this references (lines 36-45).
„The classification and grading of periodontitis is important for the classification of periodontitis. According to this, periodontitis can be initial, moderate or severe. Another classification is made according to the stages of periodontitis, where stage 1 loss of interdental clinical attachment at the site of the largest loss is 1-2 mm, in stage 2 it is 3-4 mm, in stage 3 and 4 is <5 mm . Numerous criteria have also been established for delimiting the correct classification, taking into account the systemic risks as inflammation. The most commonly used biomarker is gingival crevicular fluid.
A readjusted classification from 2018 shows the stages (1-4) and depending on the location (generalized, molar-incisor distribution), but also their degree of progression (slow rate of progression, moderate rate of progression, and rapid rate of progression).”
Comment 2.: To be included in the discussion are the new proactive approaches to reduce the incidence of inflammation through long-term probiotics, especially in diabetic patients for the reduction of bacterial load and for the modulation of glycated hemoglobin, I am attaching the reference: Professional and home-management in non-surgical periodontal therapy to evaluate the percentage of glycated hemoglobin in type 1 diabetes patients. International Journal of Clinical Dentistry, 2021, 14(1), pp. 41–53
Answer 2.:
Thank you very much for the suggestion. I will add this reference (lines 420-425).
„The link between glycosylated hemoglobin and periodontal disease was followed in a 2021 study, and a significant decrease in glycosylated hemoglobin was found in the group with rigorous oral hygiene, compared to the control group. In the current study we can follow the improvement of paraclinical parameters (Glucose, Cholesterol, Triglycerides, Uric acid, Urea nitrogen, Alkaline phosphatase, Amylase, Fructosamine, Lactate dehydrogenase).”
Reviewer 3 Report
Interesting paper on association between metabolic syndrome progression, periodontitis and its association with oral microbiome.
Paper is well written, interesting ad results are novel. I would ask for few minor additions/clarifications:
1. I would point out that metabolic syndrome is also associated with NAFLD, obesity and through shared inflammation process it is also connected to diverticular disease ( https://pubmed.ncbi.nlm.nih.gov/33951119/) and this should be described in the introduction.
2. Line 45 on page 2: In parameters for MS follow up HgbA1c is an important parameter as well. I am curious why fructosamine was mentioned instead of HgbA1c?
3. Line 89- ECW and TBC needs to be explained before abbreviation is used
4. Methods: we patients evaluated on empty stomach? the results are known to be affected depending on weather patients have eaten or are fasting
5. What was " anti-inflammatory diet" that group DG and DSG were given? How many kcal per day? protein/carb/fat ratio?
6. DSG group-please specify what was the level of physical activity and for how long
7. Author might find interesting the following, recently published paper on similar topic, it should be discussed : https://www.mdpi.com/1660-4601/18/21/11110
Author Response
Response to Reviewer 3
Firstly, I, the author of the present manuscript wish to thank you for thoughtful commentary you have provided to improve the quality of the paper. I am very grateful for the time and effort you have devoted to this task. We have extensively revised my manuscript according to the recommendations. All changes in the text and the new figures that we have redesigned are highlighted. Please, see the point-by-point answers to your comments below.
Comment 1.: I would point out that metabolic syndrome is also associated with NAFLD, obesity and through shared inflammation process it is also connected to diverticular disease ( https://pubmed.ncbi.nlm.nih.gov/33951119/) and this should be described in the introduction.
Answer 1.:
Thank you for the suggestion. The whole manuscript was extensively revised and compeleted (lines 52-58).
„Recent studies that have looked at obesity and associated diseases have shown a direct relationship between visceral fat, non-alcoholic fatty liver disease, and diverticulosis. The relationship between the intestinal microbiome and the metabolic syndrome has been followed since 2013, but dysbiosis and non-alcoholic fatty liver disease and diverticulosis appear in the most recent studies. It seems that the disease of the fatty liver, visceral fat, but also diverticulosis is the consequence of a marked dysbiosis, present in the metabolic syndrome.”
Comment 2.: Line 45 on page 2: In parameters for MS follow up HgbA1c is an important parameter as well. I am curious why fructosamine was mentioned instead of HgbA1c?
Answer 2.:
Thank you for the observation.
Fuctozamine (205-285 μmol / L) reflects the glycosylation of non-enzymatic blood proteins, and does not interfere with harvesting stress. The rate of formation of fructosamine is proportional to glucose levels, very useful for MS patients with glucose (60-100mg / dL).
Fructosamine indicates high blood sugar for 3 weeks, as well as glycosylated hemoglobin for 3 months. The evaluation of the patients took place monthly, so I could not evaluate the glycosylated hemoglobin to follow the diet therapy. For this I chose fructosamine and it is also a glycosylated protein.
Comment 3.: Line 89- ECW and TBC needs to be explained before abbreviation is used”
Answer 3.:
Thank you very much for the correction. I completed the paragraph accordingly (line 123).
Comment 4.:
Methods: we patients evaluated on empty stomach? the results are known to be affected depending on weather patients have eaten or are fasting
Answer 4.:
Thank you for the remark. I made the required corrections (lines 115-116).
„Patients were evaluated on an empty stomach in the morning.”
Comment 5.: What was " anti-inflammatory diet" that group DG and DSG were given? How many kcal per day? protein/carb/fat ratio?”
Answer 5.:
Thank you for the suggestion. I complete the manuscript with the required information (lines 101-107).
„The patients in the study followed a personalized diet, based on a healthy diet (intake of macronutrients in the percentage of 45-55% carbohydrates, 25-35% protein, and 15-20% lipids, hypocaloric, with a reduction in caloric intake by 200 kcal). The personalization consisted in testing from the venous / capillary blood a specific allergic reaction of type 3 and 4 IgG, of a number of 90 foods, foods specific to the local area and cuisine. Foods that had a specific IgG reaction were removed from the diet for 3 months, reintroduced only occasionally, until the end of the research period.”
Comment 6.: DSG group-please specify what was the level of physical activity and for how long
Answer 6.:
Gratefully accepting the observation, I complete the manuscript with the required information (lines 203-204).
„The physical activity followed was light, for cardiovascular stimulation, for 30-60 minutes, 2-3 times a week.”
Comment 7.: Author might find interesting the following, recently published paper on similar topic, it should be discussed : https://www.mdpi.com/1660-4601/18/21/11110 "
Answer 7.:
Again, I agree your suggestion, and I complete my manuscript with this discuss (lines 442-450).
„A study published in 2021, followed menopausal women in Korea, verifying the link between periodontal disease and metabolic syndrome. The association between the 2 parameters proved, but in the case of the Koreans it did not reach the significance thresholds. It can be explained that the prevalence of obesity was 33.2%. In Europe, the prevalence of obesity is 40-59.9%. Following only the adult population, due to sedentary lifestyle, this percentage is over 60%, according to WHO estimates. Thus, compared the insignificant relationship of metabolic syndrome and periodontitis in Koreans, and the significant
relationship obtained in adults in Romania, the key is the much higher rate of obesity, so it can be seen that obesity-related diseases are amplified.”
Reviewer 4 Report
In the present paper, Timea Claudia Ghitea investigated the correlation between the level of inflammation of the Metabolic Syndrome and the presence of pathogenic bacteria present in the periodontal gingival crevicular fluid, as well as the link between allopathic treatment and the presence of periodontopathogens at 3 and 6 months. The author concluded that an anti-inflammatory diet therapy contributes to the reduction of gingival inflammation and thus contributes to the reduction of the development of pathogenic bacteria in the gingival, responsible for the development of periodontal disease and directly by other chronic diseases. Overall, I think that the paper could be of interest for readers and researchers, in general. However, needs an extensive reorganization before it could be eventually published in this prestigious journal. I make some suggestions for improve the quality of the manuscript.
1) The author, if possible, should incorporate in tables the dietary pattern of the patients included in the present study (e. Mediterranean-style diet, Plants-based diet, Nordic dietary pattern, etc.); in this way, I feel that the readers can better understand the results obtained in the present clinical study and their possible application to clinical practice.
2) Please better define and discuss the power analysis of study.
3) Some medicines/drugs can be used to treat Metabolic Syndrome. This aspect could interfere with results here revealed. Please discuss this aspect and eventually take this factor into account in multiple logistic regression analysis.
4) Please better define the inclusion/exclusion criteria of Metabolic Syndrome. Generally, NCEP-ATPIII or IDF criteria are applied.
5) Recent research suggested that dietary fibers from beans, fruits and vegetables were associated with the gut microbiome composition and, accordingly, with risk of developing Metabolic Syndrome. Does the author plan to assess microbioma composition in this population? Please make a comment in the discussion section of manuscript.
6) In light of the results here obtained, please to discuss on the possible application of nutraceutics and/or antioxidants/antinflammatory compounds that, in combination with healthy diet and physical activity could provide a possible further strategy to ameliorate Metabolic Syndrome and to the reduction of gingival inflammation.
Author Response
Response to Reviewer 4
Firstly, I, the author of the present manuscript wish to thank you for thoughtful commentary you have provided to improve the quality of the paper. I am very grateful for the time and effort you have devoted to this task. We have extensively revised my manuscript according to the recommendations. All changes in the text and the new figures that we have redesigned are highlighted. Please, see the point-by-point answers to your comments below.
Comment 1.: The author, if possible, should incorporate in tables the dietary pattern of the patients included in the present study (e. Mediterranean-style diet, Plants-based diet, Nordic dietary pattern, etc.); in this way, I feel that the readers can better understand the results obtained in the present clinical study and their possible application to clinical practice.
Answer 1.:
Thank you for the suggestion. I complete the manuscript with the required information (lines 101-107).
„The patients in the study followed a personalized diet, based on a healthy diet (intake of macronutrients in the percentage of 45-55% carbohydrates, 25-35% protein, and 15-20% lipids, hypocaloric, with a reduction in caloric intake by 200 kcal). The personalization consisted in testing from the venous / capillary blood a specific allergic reaction of type 3 and 4 IgG, of a number of 90 foods, foods specific to the local area and cuisine. Foods that had a specific IgG reaction were removed from the diet for 3 months, reintroduced only occasionally, until the end of the research period.”
Comment 2.: Please better define and discuss the power analysis of study.
Answer 2.:
Thank you for the remark. I complete the manuscript with the power analysis (lines 285-287, and discuss 434-441)
„The results of this power analysis underline that a satisfactory degree of power was obtained by increasing the size of the batch, the result being 0.805 ie 80.5% for periodontal diseases at the end of the research period.”
„The adjusted R tells us how much variance in the outcome would be accounted for, if the model has been derived from the population from which the sample was taken. A 2010 study looked at the risk of developing metabolic syndrome in patients with schizophrenia, tracking genotype, and statistically processed with power analysis. The results thus received could be extrapolated to the population from which the individuals were extracted, as in our case, where the probability of developing periodontal diseases in people with metabolic syndrome was followed. The results showed a rate of 80.5% similar to the cohort studied.”
Comment 3.: Some medicines/drugs can be used to treat Metabolic Syndrome. This aspect could interfere with results here revealed. Please discuss this aspect and eventually take this factor into account in multiple logistic regression analysis.
Answer 3.:
Thank you very much for the remark. Patients with metabolic syndrome, included in the study, did not receive any drug treatment for it. The only intervention was the dietary therapy mentioned above. One of the objectives of the study is to emphasize the importance of a healthy diet for people with metabolic syndrome following the evolution without diet therapy of clinical and paraclinical parameters, and with diet therapy. I have followed and published this issue in several journals.
Comment 4.: Please better define the inclusion/exclusion criteria of Metabolic Syndrome. Generally, NCEP-ATPIII or IDF criteria are applied.
Answer 4.: Thank you for the thoughtful commentary. I complete the manuscript with the inclusion/exclusion criteria (lines 117-125).
„The diagnosis of the metabolic syndrome was made following the HOMA index, mixed dyslipidemia, hypertension, visceral fat, fat mass, but also the ratio between extracellular water and total water. The criterion for including patients in the current study was a diagnosis of metabolic syndrome of at least one year, and to present an initial periodontopathy. Moderate and severe periodontitis were the exclusion criteria. This study followed the evolution of fat mass, visceral fat, and the ratio between ECW / TBW, these having the most obvious changes following diet therapy, and they represent the highest risk for the unfavorable evolution of metabolic syndrome.”
Comment 5.: Recent research suggested that dietary fibers from beans, fruits and vegetables were associated with the gut microbiome composition and, accordingly, with risk of developing Metabolic Syndrome. Does the author plan to assess microbioma composition in this population? Please make a comment in the discussion section of manuscript.
Answer 5.: Thank you very much for the observation. I complete with the comment in the discussion section (lines 385-392).
„The microbiome has a particularly important role in the evolution of metabolic syndrome. Recent studies show a link between metabolic syndrome and inflammation and bacteria in reducing intestinal inflammation. This approach should not be missing from the management of the metabolic syndrome, especially where the provision of dietary fiber, with a role in the development and feeding of the microbiome, is insufficient. Our study, without additional intake of pre- / probiotics, can be continued with the recommendation of probiotics for amplification, being able to highlight the role of the microbiome.”
Comment 6.: In light of the results here obtained, please to discuss on the possible application of nutraceutics and/or antioxidants/antinflammatory compounds that, in combination with healthy diet and physical activity could provide a possible further strategy to ameliorate Metabolic Syndrome and to the reduction of gingival inflammation.
Answer 6.:
Gratefully accepting the suggestion. Completion in lines 393-398.
„To improve the results, recent studies suggest a possible association between Konjac glucomannan (butyrate producer), galactooligosaccharides (GOS), fructooligosaccharides (FOS), zinc citrate, cholecalciverol (vitamin D3), riboflavin (vitamin B2). Among the bacteria Akkermansia muciniphila and Faecalibacterium prausnitzii may bring additional benefits. For periodontal diseases we can obtain additional benefits with herbal infusion in oral health care.”
Round 2
Reviewer 4 Report
The author has satisfactorily responded to all my questions and made the necessary changes to the manuscript.